# New Polymeric Composites Based on Two-Dimensional Nanomaterials for Biomedical Applications

**DOI:** 10.3390/polym14071464

**Published:** 2022-04-04

**Authors:** Laura S. Pires, Fernão D. Magalhães, Artur M. Pinto

**Affiliations:** 1LEPABE, Faculdade de Engenharia, Universidade do Porto, Rua Roberto Frias, 4200-465 Porto, Portugal; ls.pires@campus.fct.unl.pt (L.S.P.); fdmagalh@fe.up.pt (F.D.M.); 2i3S—Instituto de Investigação e Inovação em Saúde, Universidade do Porto, Rua Alfredo Allen, 4200-135 Porto, Portugal; 3INEB—Instituto de Engenharia Biomédica, Universidade do Porto, Rua Alfredo Allen, 4200-135 Porto, Portugal

**Keywords:** black phosphorus, transition metal dichalcogenides, MXenes, tissue regeneration, hydrogels, scaffolds, cancer phototherapy, antimicrobial, antibacterial, drug delivery, biosensing

## Abstract

The constant evolution and advancement of the biomedical field requires robust and innovative research. Two-dimensional nanomaterials are an emerging class of materials that have risen the attention of the scientific community. Their unique properties, such as high surface-to-volume ratio, easy functionalization, photothermal conversion, among others, make them highly versatile for a plethora of applications ranging from energy storage, optoelectronics, to biomedical applications. Recent works have proven the efficiency of 2D nanomaterials for cancer photothermal therapy (PTT), drug delivery, tissue engineering, and biosensing. Combining these materials with hydrogels and scaffolds can enhance their biocompatibility and improve treatment for a variety of diseases/injuries. However, given that the use of two-dimensional nanomaterials-based polymeric composites for biomedical applications is a very recent subject, there is a lot of scattered information. Hence, this review gathers the most recent works employing these polymeric composites for biomedical applications, providing the reader with a general overview of their potential.

## 1. Introduction

After the upsurge of graphene, one of the most studied 2D nanomaterials for biomedical applications, new and emerging two-dimensional nanomaterials (2DnMat) are becoming part of a fast-growing platform aimed at their research and characterization [1,2,3,4,5]. These new materials succinctly comprise transition metal dichalcogenides (TMDs), transition metal oxides (TMOs), transition metal carbides, nitrides and carbonitrides (MXenes), black phosphorus (BP), layered double hydroxides (LDHs), 2D metal-organic frameworks (MOFs), nanoclay, hexagonal boron nitride (hBN), among others [6,7], as displayed in Figure 1. Similarly to graphene, they have unique properties that distinguish them from their bulk counterparts, such as high surface area, strong covalent bonds within the layer and weak van der Waals bonds between layers, dimensional confinement of electrons, etc. [8]. These make them suitable for a wide range of applications [9] including gas sensing [10,11], energy storage [12,13], water splitting [14,15], optoelectronics [16,17], water filtration [18,19], and diverse biomedical applications [20].

Nanomedicine, particularly paired with the use of these new 2DnMat, has tremendous potential to aid and greatly improve modern medicine in cancer theranostics, drug delivery, antimicrobial/antibacterial platforms, and tissue engineering [21], as succinctly displayed in Figure 2.

The photothermal conversion ability, which allows the conversion of light into heat [22], is one of the properties of 2DnMat that makes them attractive for biomedical applications. On top of that, the bandgap of 2DnMat is layer dependent, meaning that it can be tuned to better adjust to the desired goal and application [22,23,24]. Therefore, the photothermal conversion ability of the 2DnMat can be enhanced by changing their physicochemical properties, for example, through functionalization/doping or varying their dimensions (number of layers and lateral size). Photothermal therapy (PTT) for cancer treatment has been extensively studied. Taking advantage of the heat generated by photothermal conversion, PTT allows increasing tissue’s local temperature, causing photothermal ablation and necrosis (above 50–60 °C) or inducing cancer cell death by apoptosis at lower temperatures, making it an excellent alternative to conventional treatment methods. This procedure is non-invasive and mitigates the poor results often obtained with radiotherapy and chemotherapy’s inevitable damaging side effects [25,26,27]. The photothermal conversion ability of 2DnMat is also the property that makes them suitable for medical imaging, particularly, photoacoustic (PA) [28]. Succinctly, the PA imaging technique relies on the generation of an acoustic wave induced by the rapid tissue changes associated with the temperature increase derived from the photothermal conversion of 2DnMat. The generated waves are captured by a transducer and converted into images [28,29,30]. The use of 2DnMat for PA imaging has been successfully demonstrated [31,32].

**Figure 2 polymers-14-01464-f002:**
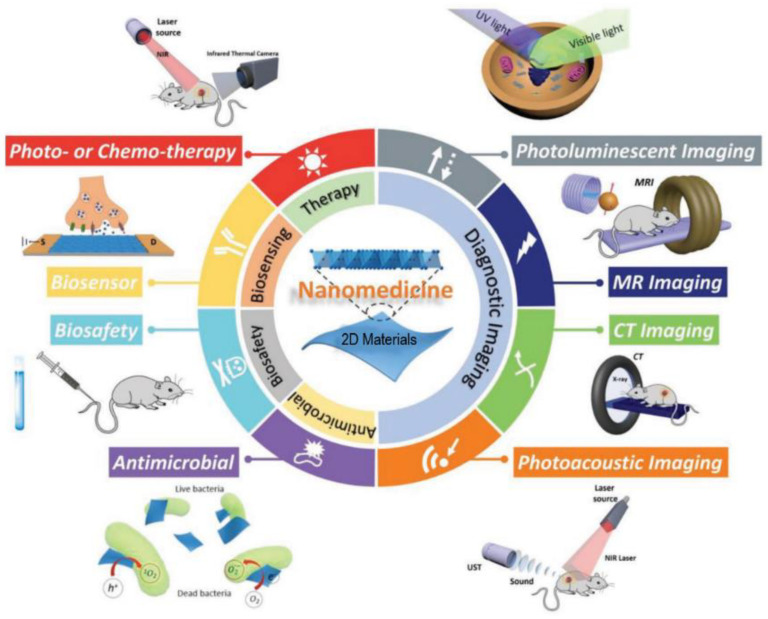
Overview of the biomedical applications of two-dimensional nanomaterials (2DnMat). 2DnMat can be used, for example, for photothermal therapy induced by near infrared (NIR) radiation; photodynamic therapy (PDT), combining light irradiation with photosensitizer drugs; for diagnostic imaging, such as photoluminescent, photoacoustic, magnetic resonance (MR) or computed tomography (CT) imaging; for antimicrobial effect through direct membrane damages or production of reactive oxygen species (ROS). Additionally,2D nanomaterials can be designed to present low cytotoxicity and adequate biosafety. They can also be used to develop innovative biosensors. Adapted from [33] © 2018 The Authors. Published by WILEY-VCH Verlag GmbH & Co. KGaA, Weinheim.

Moreover, two-dimensional nanomaterials also have inherent anti-microbial properties that can be taken advantage of in biomedical applications [34,35,36]. The anti-microbial properties can be expressed in multiple mechanisms [37,38], the most common being cell membrane disruption by the nano-knife structure characteristic of 2D materials. Antifouling properties of membranes incorporating 2DnMat have inherent advantages when compared to other membranes used for the same purpose. One of the major concerns regarding polymeric-based membranes is their poor mechanical stability and durability [39,40]. 2DnMat can increase the membrane’s mechanical properties without significantly contributing to their degradation, as they are generally inert, and provide extra resistance to chlorine [41]. Furthermore, the frequent use of silver nanoparticles due to their antibacterial properties contributes to the concerning growth of bacterial resistance [42,43]. Since the antibacterial mechanism of 2DnMat relies on their nano-knife structure, as previously mentioned in this review, the use of these materials does not contribute to this threat.

Additionally, the large surface area characteristic of 2DnMat is ideal for drug-loading, making them ideal drug delivery systems [44,45,46]. The hydrophilic nature of some or their versatility for chemical modification favors their biocompatibility [47,48]. Amongst all the methods available for surface functionalization [47,49], the main three modifications that are employed for biomedical purposes rely on polymeric-based, inorganic-based and metal-based functionalizations [6,33,50,51].

### 1.1. Examples of Biomedical Applications of 2DnMat

Sun et al. [31] demonstrated the theranostic capability of BP by using it for both photoacoustic imaging (PA) and photothermal therapy (PTT). In this study, BP was PEGylated to improve its water solubility and biocompatibility. The 2D nanomaterial demonstrated great photothermal conversion efficiency (36.8%) and photothermal stability. The group conducted in vivo assays where they intravenously injected mice with a PEGylated BP solution and PA images of the liver, kidney, and tumor pre- and post-injection were recorded. Results showed that there was a clear invasion of the particles in the major organs during the first 6 h but, at the 24 h mark, only the tumor retained signals visible in the PA images, meaning that there is a tendency for the BP particles to be retained in the tumor, whereas it can be easily excreted in the liver or kidneys. To access the PTT in vivo, mice were intratumorally injected with the PEGylated BP and exposed to NIR (near infrared) radiation, leading to a temperature increase up to 59 °C within 5 min. In 3 days, the tumors considerably shrank and scabbed, contrasting with the control groups where there was a continuous growth. Moreover, all the mice treated with PEGylated BP + NIR survived until sacrificed on day 42 whereas the rest presented an average lifespan of 18–32 days. No damage or abnormalities were observed in the histological assay.

Yin et al. [52] used MoS_2_ (molybdenum disulphide) nanosheets as a thermo-responsive drug delivery means for cancer therapy. To improve the material biocompatibility and physiological stability, the nanosheets were functionalized with chitosan (CS) and then loaded with doxorubicin (DOX), a chemotherapeutic drug. It was demonstrated that on-demand release of DOX occurred with NIR irradiation. To confirm the therapeutic potential in vivo, tumor bearing mice were intratumorally injected with the MoS_2_@CS/DOX particles and exposed to NIR radiation. It is noteworthy that the tumor temperature increased 22.5 °C, which could also induce tumor hyperthermia through PTT. Results showed that PTT combined with DOX-delivery could effectively destroy the tumor without recurrence. No histological damage or abnormalities were observed, confirming this material’s high in vivo biocompatibility and the success of the combined therapy.

Tissue engineering applications have also been explored by Zhao et al. [53] where the thermo-responsive drug release properties of MoS_2_ were employed to treat osteoarthritis (OA). In this work, the MoS_2_ nanosheets were functionalized with CS and loaded with dexamethasone (Dex). Similarly to previous reports, the 2DnMat provided an on-demand drug release profile upon NIR irradiation. In this study, a saline solution of the MoS_2_@CS/Dex (MCD) was injected directly into the joint cavity of mice suffering from arthritis. PA imaging was also recorded to infer on the residence time of Dex in the joint cavity. It was shown that the MoS_2_@CS/Dex was retained in the cavity up until 48 h, which means the MCD can have prolonged residence at the site. The inflammatory markers IL-1β, TNF-α, and IL-8 and the cartilage destruction proteins MMP13 and ADAMTS5 also had reduced expressions. The cartilage of mice submitted to MCD + NIR treatment presented ordered chondrocytes and no signs of cartilage erosion. No histological damage or abnormalities were observed in the major organs. The above mentioned are some examples of how two-dimensional nanomaterials can be applied in nanomedicine.

Overall, healthcare is increasingly evolving towards point-of-care (PoC) diagnostics and personalized medicine, translating into a growing need of more sophisticated biosensors [54]. The work of Hroncekova et al. [55] is one example of how biosensors based on 2D nanomaterials can be used in PoC. Using the MXene Ti_3_C_2_T_x_ (titanium carbide), a biosensor aimed at mitigating the high ratio of false-positive/negative results of prostate-specific antigen (PSA) detection was developed. PSA is not reliable for the early onset prostate cancer detection, therefore, an alternative marker, sarcosine, was chosen. The MXene supported SO_x_ immobilization and sarcosine was set as analyte. The developed biosensor had one of the lowest limits of detection (LOD) amongst published MXene based biosensors and presented one of the highest sensitivities aimed at detecting sarcosine.

Self-powered wearable piezoelectric nanogenerator (PENG) and triboelectric nanogenerator (TENG) have been recently studied. These devices convert the mechanical movements of the human body into energy to power the wearables. These can be employed in a plethora of applications, such as physiological signal acquisition during sports or, drug delivery, and serve as power cell for small circuits (e.g., pacemakers). This is a very vast topic by itself. Extensive reviews have been published on this subject [56,57], and the use of 2DnMat in self-powered wearables have been reported [58,59,60,61].

The application of neural implants is severely hindered by the immune response of the nervous system, particularly the inflammatory response of glial cells. Graphene has been extensively reviewed for neural interface applications, specifically focused on the interaction with glial cells, providing new grounds for the development of therapies based on neural implants [62].

### 1.2. The Current State of 2DnMat

As previously stated, graphene is one of the most studied 2D nanomaterials for biomedical applications. It has electrical conductivity, antibacterial properties, biocompatibility, and mechanical stiffness, which are all desirable aspects for various biomedical applications [63]. Its thorough investigation has unquestionably opened doors for the research of other 2D nanomaterials. Compared with graphene, MXenes tend to have higher electrical conductivities and their functional groups can enhance their mechanical properties [47]. Photodynamic therapy relies on the production of reactive oxygen species (ROS) to induce cell death [64]. Like graphene, TMDs can increase ROS levels and they have the advantage of possessing decreased cytotoxicity, alongside BP [63,65]. Tunable bandgap is one major advantage that other 2DnMat have set side-by-side to graphene, which has a bandgap of 0 eV. Although BP’s carrier mobility (1000 cm^2^/Vs) is lower than graphene, BP has increased sensitivity and precise detection limits compared with graphene-based sensors, making it a good match for biosensors based on antigen-antibody interactions [65].

Scaffolds and hydrogels are regularly used in biomedical applications including regenerative medicine, drug delivery, and antibacterial products. Some of the advantages include localized therapy, less invasive procedures, the bypassing of side effects usually associated with grafting techniques, and the overcoming of current limitations of conventional medicine, among others [66,67]. Recently, the addition of emerging 2DnMat to these scaffolds/hydrogels is beginning to attract more attention but is still poorly studied. The two-dimensional nanomaterials can further improve the mechanical properties of the scaffolds/hydrogels and enhance their interactions with cells. In some cases, they can even boost innate antibacterial properties of some of the commonly used biopolymers in these applications, like Mayerberger et al. [68] observed. Chitosan has intrinsic antibacterial properties [69], which are further improved with the addition of MXenes. The group of Mayerberger demonstrated that a nanofibrous composite of CS/MXene significantly reduced bacteria viability compared with a pristine CS scaffold.

The promising prospects of two-dimensional polymeric composites arise the need for more profound and more extensive research. There are some extensive reviews on the non-biomedical applications of polymeric composites containing two-dimensional nanomaterials, and on biomedical applications of 2DnMat alone. However, to the best of our knowledge, and despite the inherent advantages that incorporating 2DnMat in scaffolds/hydrogels brings, no paper explicitly focuses on the biomedical applications of polymeric composites based on 2DnMat. Herein, in this short review, we gathered recent studies employing 2DnMat polymeric composites in biomedical applications to further disclose their potential and possibly potentiate future investigations on this topic.

## 2. 2D Nanomaterials Polymeric Composites for Biomedical Applications

The properties that make the new two-dimensional nanomaterials suitable for biomedical applications, like high surface-to-volume ratio or photothermal conversion ability also justify their addition to the scaffolds and hydrogels already used in tissue engineering applications, among others. In fact, it has been proved that incorporating some of these materials can improve the performance of the final composite scaffolds/hydrogels [70,71].

For example, combining the photothermal properties of 2DnMat with thermo-/light responsive hydrogels can broaden the applications and goals of 2D nanomaterials in drug delivery and tissue engineering applications [22,28], by providing localized targeting [22].

Since this is a recent field, there is a lack of published studies using 2DnMat polymeric composites specifically for biomedical applications. In fact, from all of the two-dimensional nanomaterial families, most of the research papers focus mainly on using BP, TMDs and MXenes. Table 1 congregates the most recent studies containing 2DnMat-based polymeric composites for biomedical applications.

### 2.1. Black Phosphorus

After the graphene boom, black phosphorus became the new highlight of the 2DnMat family. Each layer comprises multiple corrugated planes of P atoms, each covalently bonded to three more P atoms, while the interlayer connections are made through weak van der Waals forces [103,104]. This structure gives BP great optical and mechanical properties. Even though most of the research conducted towards this material relies on optoelectronic applications [105], BP demonstrates excellent potential for use in the biomedical field, particularly theranostics, PTT, antibacterial agent and drug delivery [106].

Phosphorus is a natural occurring element in the human body, with 85% of the total being present in the bones and teeth [3,5,107]. Amongst the two-dimensional material family, BP has one of the highest biocompatibility and biodegradability. Exposure to water or oxygen, both abundantly present in the human body, easily degrades BP, which in turn produces phosphates and other P_x_O_y_. Despite the outstanding biocompatible and biodegradable properties of BP, it still remains a challenge to use this material due to its instability in ambient conditions, as the lone pair of electrons makes it undergo partial degradation even before being implanted. This can affect its performance as it greatly reduces its semiconducting properties [49,107]. However, surface modification and physical mixture greatly improve BP stability [107], allowing it to be incorporated in scaffolds and hydrogels frequently used in biomedical applications.

#### 2.1.1. PTT

The use of BP polymeric composites for photothermal therapy has been recently studied. Shao et al. [73] developed a biodegradable and photothermal sprayable platform for the post-surgical treatment of cancer as illustrated in Figure 3b, aimed at the removal of residual cancerous tissue. With that in mind, the group created a sprayable platform that, upon radiation exposure, undergoes gelation and adapts to the complex and intricate post-surgical terrain. Moreover, by incorporating the 2DnMat, which has high photothermal conversion properties, the photothermal ablation of the remaining cancerous tissue can be accomplished. BP nanosheets were combined with the thermosensitive hydrogel [poly(d,l-lactide)-poly(ethylene glycol)-poly(d,l-lactide)] (PDLLA-PEG-PDLLA: PLEL). The hydrogel gelation was not affected by the introduction of BP and the rapid biodegradable properties of the nanosheets remained intact, which is desirable for biomedical applications. The composite did not potentiate cytotoxic effects for human mesenchymal stem cells (hMSCs), L929 and HeLa cells, proving its biocompatibility in vitro. The hydrogel presents itself with good fluidity at room temperature and undergoes the sol-gel process with only 20 s of NIR (808 nm laser at 0.5 W cm^−2^) exposure (Figure 3a). To evaluate the postulated purpose of this hydrogel composite in vivo, tumor models were established on Balb/c mice with in situ subcutaneous injections of HeLa cells. The post-operated area is then irradiated with NIR radiation, and the hydrogel composite is sprayed onto the wound region. Figure 3c shows that within 30 s, the temperature reaches 58.2 °C, which ensures tissue ablation. The mice submitted to the BP/PLEL hydrogel + NIR treatment were completely cured within 16 days with no recurrence, contrasting with the control group that had an 80% recurrence rate after 8 days post-surgery (Figure 3d). The life span was also severely affected. The no-treatment group had a life span of 18–24 days, the surgery-only group lived for 35 days and the surgery + BP/PLEL + NIR survived for over 2 months. No histological damage or abnormalities were observed for the BP/PLEL + NIR mice. Bacterial infections are always a serious concern after any surgical procedure. With that in mind, the antibacterial properties of the BP/PLEL composite were also studied. The results showed that combining the composite with 1 min NIR irradiation reduced the *S. aureus* population by 99.5%, giving the composite the ability to eliminate residual tumor tissue and bacteria simultaneously.

Similarly, Xing et al. [74] used a BP/cellulose hydrogel composite for cancer treatment, also taking advantage of the photothermal properties of BP. The incorporation of BP did not compromise the structure of the hydrogel, in fact, the elastic modulus was enhanced as well as its ability to retain water upon stress. The BP/cellulose composite presented a good photothermal cycle stability and was able to reach a 50 °C temperature after 600 s of NIR irradiation (808 nm, laser 1 W cm^−2^). In vitro and in vivo cytotoxicity assays were conducted using B16, SMMC-7721 and J774.1 cells and Balb/c nude mice subcutaneously injected with the composite hydrogel. No cytotoxicity was observed, and the mice remained healthy. The histological examination also revealed no signs of damage to the main organs nor increased inflammatory cytokine levels (TNF-α, IL-1β, and IL-6), confirming biocompatibility and biosafety for living organisms. In vitro PTT assessment was also conducted using the same cell lines, obtaining an almost 100% killing cell efficiency for 10 min of radiation exposure. This was followed by the application of PTT therapy in vivo by subcutaneous implantation of the composite after establishing tumor models by subcutaneous injection of SMMC-7721 cells. The mice treated with the combination of BP/Cellulose + NIR displayed a significant tumor regression while only a slight decrease in tumor volume or no change at all was observed in the control groups.

#### 2.1.2. Drug Delivery

Due to the high surface ratio of BP nanosheets, drug delivery via BP polymeric composites has also been researched. By using thermo-responsive hydrogels, an on-demand drug release can be obtained caused by the heat generated from the BP exposure to NIR radiation. Yang et al. [72] produced a composite hydrogel from polyvinyl alcohol (PVA) and surface-modified BP with poly(dopamine) (PDA)—pBP. The mechanical properties of the hydrogel were significantly improved with the addition of pBP, having a 90% increase of elongation at break. The composite presented good photothermal stability and the NIR heat conversion efficiency was calculated at 31.5%. Congo red was chosen as a drug model. The composite’s drug loading level (LL) increased with the addition of pBP, as expected, going from 80% to ~95%. The induced drug release was studied by irradiating the drug-loaded composite with NIR radiation (808 nm, 2 W, on/off cycles). An abrupt release was observed for each on-cycle, resulting in a stepwise drug release profile. The controllable drug delivery stems from the destruction of the hydrogel hydrogen bonds induced by the generated heat of the pBP + NIR. The biocompatibility of this PVA/pBP composite was studied using 3T3 fibroblasts, with no cytotoxic effects observed.

The same radiation-responsive drug release concept was studied by Qiu et al. [75], this time using a DOX-loaded agarose hydrogel with PEGylated BP. Similar to the previously reported work, the composite presented good photothermal stability and a photothermal efficiency of 38.8% with NIR (808 nm, 1 W cm^−2^). Complementing Yang’s research, Qiu’s group performed both in vitro and in vivo assays to confirm the thermo-responsive drug release properties of the composite. No cytotoxic effects were observed for MDA-MB-231, A549, HeLa, and B16 cells. Furthermore, MDA-MB-231 cell death derived from the controlled release of DOX upon irradiation confirms the therapeutic potential of the composite in vitro. In vivo tests were carried out in tumor-bearing mice by injecting the Agarose/BP@DOX hydrogel intratumorally. The group treated with the composite, combined with NIR-induced drug release, presented significant tumor reduction. No damage or abnormalities were observed in the histological assessment of the major organs.

#### 2.1.3. Wound Healing

Another interesting approach for using polymeric composites with BP is wound healing. Particularly, diabetic ulcer treatment still presents a big challenge due to the frequent bacterial infections and exacerbated inflammation at the site. This is where the work of Ouyang et al. [77] was focused. Aimed at treating diabetic ulcers, they produced a sprayable analgesic fibrin gel incorporated with BP nanosheets, suitable for wound healing and drug delivery. In vitro assays with human umbilical vein endothelial cells (HUVECs) demonstrated increased proliferation rates and cell differentiation, which are essential for wound healing. One of the setbacks that can hinder the healing process of open wounds is bacterial infections. Due to the photothermal conversion ability of BP, exposing the composite to NIR radiation significantly increased the temperature of the gel, which in turn reduced the bacterial population of *S. aureus* by ~94.3%. Continuing with NIR irradiation, the analgesic drug release lidocaine (Lic), induced by the irradiation was studied in vivo. Although fibrin is not a thermo-responsive gel, the increase of temperature due to the photothermal conversion properties of BP can induce drug release by the loosened structure of the gel upon high temperatures. Using a von Frey test, the results showed that the anesthetic effects of lidocaine were substantially longer when using the localized delivery than administering free lidocaine. Further in vivo assays were conducted on diabetic mice with infected wounds. Greatly improved wound healing was observed on the groups treated with composite + Lic + NIR. This group had the wound reduced to 50% of the original size in 5.7 days, while the control took 9.3 days.

#### 2.1.4. Tissue Engineering

Due to the degradation of BP into P-derivatives, the inherent compatibility of BP with bone tissue engineering has also gained interest in the scientific community. The use of a BP polymeric composite for the treatment of osteosarcoma, an aggressive bone cancer, has been reported by Yang et al. [78]. The group produced a 3D printed scaffold based on a bioglass/BP (BG/BP) polymeric composite. Bioglass (BG) not only serves as a matrix for BP support but also actively participates in bone regeneration. Furthermore, by having the BP degrade into P_x_O_y_ products, the formation of calcium phosphate is stimulated. As expected, the photothermal conversion increases with the addition of BP to the 3D-printed scaffold, having an increase of 36.3 °C upon 5 min of NIR exposure, contrasting with the 3.6 °C increase of pristine BG. The composite promoted elevated values of Saos-2 cell proliferation, demonstrating its high biocompatibility. In situ biomineralization was also evaluated for BP nanosheets by submerging them in simulated body fluid (SBF). On day 8, most of the nanosheets had degraded and transformed into calcium phosphate (CAP), evidencing the suitability of BP for bone tissue engineering. The 3D-printed composite scaffolds were implanted on tumor-bearing mice and subsequently submitted to NIR. In 5 min of irradiation, the tumor temperature rose to 58 °C, completely eliminating the tumor without recurrence. Histological analysis showed no damage or abnormalities to the main organs. The osteoinductive properties of the composite was demonstrated in vitro using human bone marrow stromal cells (hBMSCs). On day 5, there were clear signs of osteogenic differentiation by change of cell phenotype. The same process was observed in pristine BG scaffolds, but it progressed at a much slower pace. Osteogenic gene expression was also evaluated by controlling the OPN, OCN, ALP, COL 1, and RUNX 2 markers. The group performed in vivo assays where a cranial defect model was induced in Sprague-Dawley rats. The composite 3D-printed scaffolds were then implanted and induced osseous tissue formation.

Huang et al. [76] used a BP/arginine-based unsaturated poly(ester amide)/gelatin methacrylamide hydrogel (BP/PEA/GelMA hydrogel) composite to enhance bone regeneration. Like Yang’s group, Huang et al. took advantage of the bi-products of BP degradation to promote the formation of new calcium phosphate. Similarly, the hydrogel stimulated osteogenic differentiation of human dental pulp stem cells (hDPSCs). This time, in vivo tests were carried out in rabbits where newly formed vessels were detectable in 4 weeks post-implantation, and at week 12, the bone was completely repaired.

Quian et al. [80] studied, for the first time, the use of BP for nerve regeneration. The group produced an implantable BP/polycaprolactone (PCL) nanoscaffold. A severe neurological defect model in Sprague−Dawley rats was used for the in vivo testing. No long-term toxicity was observed 6 months post-surgery, demonstrating the biosafety of BP. Compared with the control group where the rats were treated with an autograft, the vascular endothelial growth factor (VEGF) and the vascular marker CD34 presented increased expressions on the group treated with the BP/PCL nanoscaffold. Furthermore, the neurofilament 200 (NF200) and β-III-tubulin (Tuj1) also had higher expressions in the group treated with the composite, which confirms successful nerve regeneration.

### 2.2. TMDs

TMDs have a MX_2_ stoichiometry (M = transition metal; X = chalcogen) in the form of X-M-X. Similarly to BP, on each layer, the M and X are bonded through strong covalent bonds and the various layers are held together through weak van der Waals forces [108,109].

The use of 2D TMDs in polymeric composites has not been thoroughly explored. However, the two published works demonstrate promising prospects. In these two studies, two-dimensional MoS_2_ was used. Wu et al. [82] studied the effect of a polyacrylonitrile/ MoS_2_ (PAN/MoS_2_) electrospun scaffold on bone marrow mesenchymal stem cells (BMSCs). With the addition of the TMD, the nanofibers started to present rougher surfaces with needle-like structures. BMSCs were cultured on the composite scaffolds for a 7-day period, while their cytotoxicity and proliferation were evaluated. After 7 days, there was no significant difference between the cell viability of the control and composite groups, demonstrating the excellent biocompatibility of the composite. Moreover, the composite with the highest TMD concentration (40%) significantly increased osteogenic differentiation by day 14, determined by the measurement of alkaline phosphatase (ALP) activity. These results corroborate the use of this material in tissue engineering applications.

Using the same TMD, an injectable implant for drug delivery and tumor hyperthermia was developed by Wang et al. [81]. The composite implant was made from poly(lactic-co-glycolic acid) (PLGA), DOX, and PEGylated MoS_2_, all dispersed in N-methylpyrrolidone (NMP) to form an injectable oleosol that solidifies when in contact with water (PLGA/MoS_2_@PEG/DOX), as illustrated in Figure 4a. The photothermal properties were evaluated by irradiating the composite with NIR radiation, and after 5 min, the implant temperature increased 50 °C. The composite did not reveal any cytotoxic effects towards L929 cells and in vitro assays showed no increased blood coagulation derived from the implant. The implant also demonstrated a very high drug loading capacity, with 95% of DOX loaded when a 1 g/mL PLGA concentration was used. The drug release profile was slow without NIR, despite the tumors’ natural acidic environment slightly accelerating the delivery. However, a controlled drug release profile was achieved by exposing the loaded implant to NIR radiation, see Figure 4c). Results showed that upon 5 min of NIR, the drug release increased from 8.7% to 31.8%. The evaluation of in vivo performance was also conducted on tumor-bearing mice. After injection, the implant solidified and did not leak, ensuring a localized therapy. NIR irradiation could elevate the tumor temperature beyond the therapeutic threshold, validating the PTT ability for tumor ablation/hyperthermia (Figure 4b). Only the combined treatment of implant + NIR shrank the tumor, which almost disappeared by day 7. Without NIR radiation, the tumor volume increased 507.1% after 28 days, as displayed in Figure 4d, whereas with NIR, the tumor was successfully irradicated, and the scar was completely healed. The histological analysis showed that less than 1% of MoS_2_ was present in the major organs. Furthermore, no apparent damage or abnormalities were observed in the organs of the mice.

### 2.3. MXenes

MXenes are a newly discovered class of 2DnMat with a M_n+1_X_n_T_x_ structure (M = transition metal; X = C or N and T_x_ = surface groups) [110,111]. The functional groups of MXenes make them have higher hydrophilicity and capable of easy surface modification [22]. To this date, MXenes have been primarily studied for energy storage applications. However, there has been some attention directed towards their use for biomedical applications, particularly Ti_3_C_2_, considering that Ti is an overall inert metal, and C and N are elements that compose all life forms [5,112,113,114]. The graphene analog borophene is a rising Mxene that could have future applications in the biomedical field. Since borophene is a new material, very few research has been conducted on the use of this 2DnMat for biomedical applications. However, it is known that borophene has NIR absorption properties suitable for PTT, high surface-to-volume capable of drug-loading, it can generate ROS upon irradiation, and demonstrates promising prospects for biosensing applications. Like borophene, other graphene analogs such as germanene and silicene have similar future perspectives [115,116]. The extremely limited research on the use of MXenes for biomedical applications beyond Ti_3_C_2_ and the very restricted portfolio outside cancer treatment through PTT showcases the need for a more comprehensive investigation. With that in mind, Lin et al. [117] studied the use of Ta_4_C_3_ for PTT + imaging. The MXene choice was based on Ta having a higher atomic number, making it have higher contrast for CT imaging and the exceptional photothermal conversion of Ta_4_C_3_.

#### 2.3.1. In Vitro Studies of Mxene-Based Polymeric Composites

The use of MXenes, especially Ti_3_C_2_/Ti_3_C_2_T_x_ polymeric composites for biomedical applications, has been highly studied compared with other 2D nanomaterials. Awasthi et al. [93] produced and studied electrospun polycaprolactone/MXene (PCL/MXene) composite. With the incorporation of the MXene, the average fiber’s diameter increased, presumably due to the interconnected MXene along the fibers. In in vitro biomineralization studies, it was observed that higher ratios of hydroxyapatite crystals formed in the fibrous composite. The heightened calcium and phosphate ion deposition was thought to be due to the increased wettability of the composite scaffolds. Further in vitro studies were conducted, this time evaluating the biocompatibility of the PCL/MXene fibers using two different cell lines, pre-osteoblast MC3T3-E1 and fibroblast NIH-3T3, presenting a cell viability of ~72% and ~70%, respectively. The good bioactivity of the composite fibers aligned with adequate fiber/cell interaction and exhibit the potential of these composite fibers for biomedical applications.

Rongkang et al. [96] also studied the future use of poly-L-lactic acid-polyhydroxyalkanoates/MXene (PLLA-PHA/MXene) composite polymeric nanofibers for tissue engineering applications. Using BMSCs, the group observed higher adhesion and slightly elevated proliferation rates on the composite nanofibers. After 5 days of culture, the cells grew on both scaffolds (with and without the MXene). However, on the pristine PLLA-PHA nanofibers, the cells presented a contracted state, whereas an integrated cell-fiber growth was observed in the composite. This distinct morphology might be due to the MXene nanosheets providing a good environment for cell activity. Moreover, the osteogenic performance was evaluated by PCR for COL1a1, OCN, OPN and RUNX2. On day 14, the cells seeded on the composite nanofibers presented a much better osteogenic differentiation than the pristine PLLA-PHA fibers.

Another study directed at bone regeneration using a poly(lactic acid)/MXene/n-octyltriethoxysilane (PLA/MXene@OTES) membrane was conducted by Ke Chen et al. [83]. The collected outcomes are aligned with what was already reported by other groups. The mechanical properties of the membrane increased with the addition the MXene. In vitro assays using MC3T3-E1 cells demonstrated increased adhesion, proliferation, and osteogenic differentiation for composite membranes.

An unconventional composite hydrogel using honey, chitosan, and MXene nanosheets was created by Alireza et al. [91]. This novel choice of materials was based on honey’s anti-inflammatory, antibacterial, and antioxidant properties. The addition of MXene did not affect the structure of the hydrogel. The composite hydrogel demonstrated good swelling ability, biodegradable properties, electrically conductive behavior, and self-healing abilities. No cytotoxic effects were observed for mesenchymal stem cells (MSCs) and human induced pluripotent stem cells derived cardiomyocytes (iPSCs) cells. The group also stated how adding honey and MXene to chitosan could improve its application in tissue engineering applications and cancer.

#### 2.3.2. PTT

The photothermal properties of Ti_3_C_2_(T_x_) have also been investigated to be employed in PTT of cancer. Pan et al. [92] incorporated MXene in a BG 3D-printed scaffold for cancer treatment while accelerating tissue reconstruction. The composite scaffold displayed good PTT ability when irradiated with NIR (808 nm, 1 W cm^−2^). No cytotoxic effects were observed for Saos-2 cells. In vivo studies were carried out in mice bearing Saos-2 bone tumors. After the scaffold implantation, NIR irradiation elevated the tumor temperature up to 63 °C in about 2 min, whereas the tumor temperature only increased to about 37 °C when using pristine BG. The BG/MXene hydrogel did not induce histological changes in the major organs. The osteogenic capability of the composite scaffold was verified using hBMSCs. The cells both adhered and proliferated at higher rates on the composite scaffold. Furthermore, the gene expression of COL I, RUNX2, OCN, and OPN increased, see Figure 5a–d, meaning that the composite promotes cell differentiation. Compared with pristine BG scaffolds, the BG/Ti_3_C_2_ 3D-printed scaffolds promoted increased bone regeneration in Sprague–Dawley rats with critical cranial defects, observed by higher tissue calcification (Figure 5e).

On a further note, Chenyang Xing et al. [94] used the photothermal properties of the MXene and studied the possibility of having a light/thermo-responsive drug release profile while also performing PTT. With that in mind, a cellulose/MXene loaded with DOX was produced. Incorporating the MXene in the cellulose hydrogel did not significantly change the amount of trapped water within the cellulose chains, which is favorable for drug loading. However, the loading capacity of the composite hydrogel remained roughly the same, indicating that the incorporation of MXene did not significantly change the microstructure of the hydrogel. Without NIR irradiation, the drug release in water is slow, while the drug release is promptly accelerated with NIR irradiation. No cytotoxic effects were observed for HepAl-6, SMMC-7721, HepG2, U-118MG, and U-251MG cells. NIR irradiation for an MXene concentration of 235.2 ppm led to an almost 100% killing efficiency of tumor cells. Tumor-bearing mice intratumorally injected with the composite did not present histological changes in the major organs. Additionally, the pro-inflammatory TNF-α, IL-6, and IL-1β cytokine levels were also measured with no significant changes, demonstrating good biocompatibility in vivo. Dual modal PTT/chemotherapy was successful in vivo as it completely eliminated tumor cells with no recurrence.

#### 2.3.3. Drug Delivery

MXene polymeric composites have also been researched for drug delivery, starting with the work conducted by Zhang et al. [95]. They produced a polyacrylamide/Ti_3_C_2_ (PAM/Ti_3_C_2_) hydrogel and studied its mechanical and drug release properties. The incorporation of the MXene improved the mechanical behavior of the hydrogel yielding in a higher elongation, compression, bending degrees, knotting without breaking, and no fractures were observed under stress. Additionally, the MXene nanosheets greatly improved the swelling ability of the PAM hydrogel. The group used chloramphenicol as a model drug to evaluate the drug release profile. As expected, higher drug loads were observed on the composite hydrogel. The release effect was evaluated in a hydrochloric acid (HCl) solution of 1.2 pH at 37 °C. The composite exhibited higher drug loads and with higher drug release, allied with enhanced mechanical properties, the potential for applications in biomedicine was confirmed.

#### 2.3.4. Wound Healing

Taking advantage of the photothermal conversion of MXenes, Jin et al. [97] produced fibrous nanobelts composites from polyacrylonitrile-polyvinylpyrrolidone/Ti_3_C_2_@ P(AAm-co-AN-CO-VIm) (PAN-PVP/Ti_3_C_2_@PAAV). The composites are NIR-responsive and aimed at wound healing through the controlled release of vitamin E. The nanobelt fibers’ structure is composed of PAN and PVP. The incorporated MXene was functionalized with a thermo-responsive polymeric layer, PAAV. The composites showed good photothermal stability, with and without the PAAV coating. For the purpose of this work, there was no need for tissue hyperthermia, therefore, a lower power of 0.33 W was chosen for NIR irradiation. In 3 min, the temperature reached 41 °C. The excellent biocompatibility of the composite was verified through the enhanced cell adhesion and proliferation of BMSCs. Not only does the composite provide a good spatial structure for the cells, but it also supplies them with a suitable environment for cell growth and proliferation, observed by the integrated cell fiber structure obtained in the BMSCs cultured on the PAN-PVP/ Ti_3_C_2_@PAAV composite. Given the exceptional biocompatibility in vitro, further in vivo studies with the composite loaded with vitamin E were conducted in mice. The thermo-responsive PAAV layer relaxes the structure and vitamin E is released. The wounds on mice treated with the composite + vitamin E + NIR contracted much faster than the other groups. It is noteworthy that the newly epithelial tissue was much thicker in the composite with the PAAV coating, demonstrating its efficiency in vitamin E release.

A different approach using the same methodology was explored by Xu et al. [98], where PVA/AMX/Ti_3_C_2_ Nanofibrous Membrane was produced for wound healing. The polymeric composite had amoxicillin (AMX)—an antibiotic—incorporated that was released by NIR exposure. The composite had good photothermal stability. The temperature change of the scaffold under NIR (808 nm laser) was monitored to infer the thermal stability of the membrane. Moreover, the photothermal conversion efficiency was calculated at 41.9%, higher than the currently used conventional photothermal agents. In vitro AMX release actively increases with NIR irradiation, which allows for a controlled release. The antibacterial ability of the composite was accessed to confirm the composite suitability for wound healing applications. For a 24 h incubation period of *E. coli*, the antibacterial rates of 84.2% and 96.1% were obtained without and with NIR irradiation, respectively. Under the same conditions, an antibacterial rate of 99.1% was observed *S. aureus*. In vitro phototoxicity, hemocompatibility and cytocompatibility were evaluated using L929 cells. Cell viability above 90% was obtained for the cells co-cultured with the composite, and the hemocompatibility was deemed enough for proceeding to in vivo studies. Balb/c mice infected with *S. aureus* were used to perform the in vivo assays. Faster wound healing rates were obtained in mice treated with the membrane and NIR irradiation combination. No histological or hemo-toxicity effects were observed, and inflammation was also neglectable in mice treated with the composite membrane + NIR.

Still on the same wound healing topic, Zhou et al. [87] created a MXene@poly(dopamine)/poly(glycerol-ethylenimine)/oxidized hyaluronic acid (MXene@PDA/PGE/HCHO) multifunctional wound healing and antibacterial scaffold, particularly aimed at destroying the multidrug-resistant bacteria *MRSA*. The mechanical properties of the scaffold increased with the addition of MXene@PDA. The composite scaffold also presented good self-healing abilities, confirmed by the quick recovery of two pieces that were cut and then placed together for 2 min. Furthermore, no significant changes on the behavior of the scaffold were observed after the self-healing process. The electrical conductivity was also higher, as expected from the addition of the MXene. L929 cells adhered to the composite and did not undergo an obvious change in nuclear shape. The cell viability exceeded 99% after a 24 h incubation period, demonstrating the excellent biocompatibility of the MXene@PDA/PGE/HCHO composite. The antibacterial properties of the composite scaffolds were evaluated with a 98.6% inhibition of bacteria growth for *E. coli*, 99.9% for *S. aureus*, and 99.03% for *MRSA*. A mouse hemorrhaging liver model and tail bleeding time assays were carried out to infer the performance in vivo. The treatment with the composite was able to control the bleeding much faster, displayed by an estimated blood loss of 24.4 ± 5.39 mg, contrasting with the control group, where it was determined at 155.6 ± 7.63 mg after 60 s. The in vivo tail bleeding assay confirmed the previous results with a significant reduction if blood coagulation time. No histological damage was observed in the major organs post-injection. After 7 days, almost no Ti was detected in the mice’s organs, suggesting that the MXene@PDA was completely degraded. Facilitated wound healing was also observed by increased gene expression of α-actin, COL III, and Vascular endothelial growth factor (VEGF). To confirm the multifunctional ability of the produced scaffold, the group used a full-thickness *MRSA*-infected wound model. On day 7, the wound area of the group treated with the composite scaffold was significantly reduced, and on day 14, the wound was covered in newly formed skin with a 96.31% closure ratio. On day 3, a decrease of the pro-inflammatory IL-6 cytokine expression was observed in the groups treated with the composite scaffold, contrasting with the control groups where a severe inflammation response was present. Compared with other groups, the MXene@PDA/PGE/HCHO scaffold effectively stimulated the healing of *MRSA* infected wounds by eliminating the bacteria and reducing inflammation.

Another multifunctional scaffold aimed at infection-impaired skin therapy was studied by Zheng et al. [86]. They produced a pluronic F127-polyethylenimine-oxidized sodium alginate/Ti_3_C_2_T_x_@CeO_2_ (F127-PEI-OSA/Ti_3_C_2_T_x_@CeO_2_) hydrogel. The use of CeO_2_ was an attempt to scavenge the reactive oxygen species (ROS) that contribute to wound inflammation. The thermosensitive and self-healing properties were evaluated with increased performance with the addition of the MXene. Additionally, outstanding adhesion and injectability were observed. The 68% weight retention demonstrated the stability of the scaffolds after being immersed in phosphate-buffered saline (PBS) for 25 days. Anew, no cytotoxic effects were observed for L929 cells and in vitro studies showed that the addition of Ti_3_C_2_T_x_@CeO_2_ successfully decreased the presence of ROS. Since the MXene increases the conductivity of the hydrogel, the proliferation of fibroblasts was prompted by a 600 mV electrical stimulation (ES). The scratch assays demonstrated that within 24 h, the composite + ES reduced the unhealed portion by 73.6%, confirming, once again, the remarkable self-healing properties of the hydrogel. As expected from previous works, the incorporation of Ti_3_C_2_T_x_@CeO_2_ enhances the antibacterial properties of the F127-PEI-OSA hydrogel, with an almost 100% reduction of the bacterial colonies (*E. coli*, *S. aureus*, and *MRSA*). The disruption of the negatively charged bacterial cell membrane caused by the attraction to the cationic PEI combined with the nano-knife structure of the MXene that breaks the cell integrity are what confer the composite hydrogel its extremely efficient antibacterial properties. In vivo histological assays showed no toxic effects or abnormalities. A full-thickness *MRSA*-infected mouse model was used to evaluate anti-infection and wound healing abilities in vivo. Like in vitro assays, the hydrogel composite could clear almost 100% of the bacterial colonies by day 14. The wounds treated with the hydrogel presented a dramatically accelerated wound healing process. On day 7, the mice treated with the composite had a 60.9% wound closure, while the control group had a closure ratio as low as 28.9%. Anti-inflammatory cytokines were also higher in the subjects treated with the hydrogel composite. There was a clear early on-set angiogenesis with treatments made with the composite, evaluated through fluorescence microscopy. The composite demonstrated superior antibacterial ability and wound healing recovery in vivo. The antibacterial properties against *S. aureus* bacteria of a PLA/Mxene composite nanofibrous scaffold were also studied by Kyrylenko et al. [85]. No cytotoxic effects were observed with U2OS cells, and the adhesion of *S. aureus* bacteria was decreased with the composite nanofibers.

#### 2.3.5. Biosensing

One of the applications in the biomedical field that is poorly studied with 2DnMat polymeric composites is biosensing. MXenes could possibly replace graphene-based biosensors due to their greater dispersibility in aqueous environments, higher electrical conductivities, and enhanced self-healable inducing properties [89]. We found three current studies that employ these materials aimed at wearable sensors to detect human biological activities. The first one was a work conducted by Liao et al. [88], where a polyacrylamide-polyvinyl alcohol/Ti_3_C_2_T_x_ (PAAm-PVA/Ti_3_C_2_T_x_) organohydrogel was produced. The organohydrogel displayed remarkable anti-freezing properties up to −40 °C, accomplished by the partial replacement of water with ethylene glycol (EG) (Figure 6a); long-lasting moisture retention (90% retained in 8 days); a striking 85% self-heling efficiency just after healing for 12 h (Figure 6c,d); and great mechanical properties, up to 350% strain and 980% deformation. The performance of the organohydrogel was maintained at subzero temperatures (Figure 6b). It retained its 3D structure with the addition of the MXene and increased its electrical conductivity. The sensor could detect real-time finger and swallowing motions (Figure 6e), even at extreme temperatures. Electronic skins, human-machine interactions, and personalized health monitoring are possible future applications of these sensors.

Zhang et al. [89] also produced a self-healable MXene polymeric composite hydrogel for wearable electronic skin applications, this time using PVA. The 3D structure of the hydrogel from the crosslinking process presumably enhances the stretchability and response. The hydrogel also demonstrated significant self-healing properties; after 5 cycles of cutting, the self-healing ability remains at 97.4%. To demonstrate the feasibility of the hydrogel for electronic skin applications, the sensor was tested by sensing the swallowing movements of the throat. It was consistently sensed throughout the several tests conducted. Finger movements were also recorded and, to prove the self-healing ability, the results were compared with a healed hydrogel that was previously cut. The performance of the device was maintained. Lastly, Sharma et al. [90] also developed a wearable sensor based on a poly(vinylidene) fluoride-trifluoroethylene/Ti_3_C_2_T_x_ (PVDF-TrFE/Ti_3_C_2_T_x_) nanofibrous composite that could detect and recognize pulse signals in the wrist and breathing motions. The sensor also demonstrated promising results for future aiding of Parkinson’s diagnosis by measuring unnoticeable resting tremors in hands. A MXene-based electrode for neural applications has been developed by Driscoll et al. [118]. The group micropatterned Ti_3_C_2_T_x_ onto a flexible parylene-C substrate and, compared with other devices, presented sensitive detection of neuronal spiking activity.

#### 2.3.6. Gas Therapy

It is evident that most of the MXene polymeric composite research is directed towards Ti_3_C_2_(T_x_), however, a recent study conducted by Yang et al. [99] used the MXene Nb_2_C (niobium carbide) coated with mesoporous silica and loaded with S-nitrosothiol (R-SNO) as the 2D nanomaterial in a BG 3D-printed scaffold for a novel nitric oxide (NO)-augmented bone regeneration. In a controlled way triggered by NIR radiation, the goal was to release a profile of high and low concentrations of NO. Firstly, high concentrations of NO would act on its anti-cancer properties (Figure 7a) and afterward, low NO concentrations would promote angiogenesis and bone regeneration (Figure 7b), taking advantage of the dual functionality of NO. Due to the photothermal conversion of Nb_2_C, the scaffold temperature reached 52 °C and demonstrated good photothermal stability upon irradiation. The NO release is triggered by radiation exposure, maintaining a slow spontaneous release afterward. In vitro assays were performed with Saos-2 cells. A cell viability of almost 100% was observed in all groups. After 10 min of NIR-II exposure, the cell viability decreased to 25–30% due to the heat generated combined with the release of NO. To evaluate tumor ablation in vivo, an ectopic osteosarcoma mice model was used, and a cranial defect model was used for bone regeneration assessment. The combination of NO release and photothermal therapy yielded better results on both tests. Not only was the scaffold able to treat tumor-bearing mice, but it also revealed excellent osteogenic and angiogenic performance. No damage or abnormalities were observed in the histological assessment.

### 2.4. Other 2D Nanomaterials

Other emerging two-dimensional nanomaterials that come from the BP family include SiP (silicon phosphide) and GeP (germanium phosphide). These share the same biodegradable and biocompatible properties that make BP so desirable for biomedical applications. Furthermore, they also have higher photothermal conversion efficiencies, which makes them excellent candidates for PTT [119,120,121,122]. The use of a SiP polymeric composite for bone regeneration has been studied by Xu et al. [100], where a GelMA- Poly(ethylene glycol) diacrylate/SiP@acryloyl chloride (GelMA-PEGDA/SiP@AC) 3D-printed hydrogel was produced. To integrate the SiP nanosheets into the GelMA-PEGDA hydrogel, the SiP was functionalized with acryloyl chloride (AC). The composite 3D-printed hydrogel scaffolds had similar rheological and mechanical properties compared with the controls. The swelling ratio also remained similar. The composite had a 235% while the pristine hydrogel 259%. It was also observed that both Si and P have a slow release, but P ions have a faster release than Si ions. This was attributed to the lone pair of P electrons, which makes it more reactive towards oxygen molecules. For concentrations up to 0.5% SiP@AC, no cytotoxic effects were observed, however, for concentrations above, there is an obvious cytotoxic response. The proliferation rate of BMSCs was similar for all groups, all presenting excellent cytocompatibility levels. The osteogenic marker ALP was the highest and ARS staining showed more mineralized nodules in the composite hydrogel. Higher osteogenic differentiation markers (Opn, Runx2 and Col-I) were also observed in the composite. The angiogenic genes (VEGF and basic fibroblast growth factor (bFGF)) had higher expressions for the composite. In vivo angiogenic and osteogenic tests were performed on mice with a 5 mm calvarial defect model. Bone growth was the highest for the SiP@AC hydrogel and no histological toxicity was observed. Furthermore, the composite promoted vascularized bone regeneration.

The same author also researched the potential of using a hyaluronic acid-graft-dopamine (HA-DA)/GeP@PDA HA-DA/GeP@PDA injectable hydrogel for spinal cord injury [101]. Dopamine (DA) was grafted onto hyaluronic acid (HA) to ensure good tissue adhesion and the GeP nanosheets were functionalized with PDA to improve its biostability and compatibility. Incorporating GeP@PDA in the hydrogel reduced its swelling ratio but the conductivity significantly increased from 0.054 S/m to 0.365 S/m. Concentrations under 0.5% of GeP@PDA did not induce cytotoxicity to neural stem cells (NSCs). The NSCs adhered well to the composite, promoting neurite outgrowth and an elongated shape, characteristic of nerve cell differentiation. Furthermore, the neurons cultured on the composite presented a higher number of connected dendrites with other neurons, forming a network. This differentiation was confirmed by confocal microscopy images of the immunostained Nestin, Tuj1, MAP2, and GFAP of the NSCs cultured on HA-DA and HA-DA/GeP@PDA was studied. In vivo studies were performed on mice with a spinal cord injury model. The implantation of the composite hydrogel greatly enhanced coordination movements and functional recovery. The distinctive spinal cord cavity associated with spinal cord injury was smaller in mice treated with the composite scaffold. Immunohistofluorescence and gene expression indicated elevated levels of the anti-inflammatory factors IL-10 and low TNF-α. Therefore, the hydrogel has an immunomodulatory role, possibly due to the phosphorus-based material’s ROS scavenging ability. 6 weeks post-surgery, there were elevated levels of CD31-labeled vascular endothelial cells and neovascularization VEGF in the composite hydrogel. The composite reduced the invasion of the inflammatory response around the lesion area and enhanced the secretion of vascular endothelial growth factor, which translated into enhanced angiogenesis. The increased angiogenesis was credited to the presence of the P element released from the GeP nanosheets. The neuron-specific marker proteins (Tuj-1, GFAP, NF 200, and MAP-2) from the lesion site were immunostained to evaluate the nerve tissue regeneration. All of them, except GFAP, had higher values, which confirms the ability of the composite for spinal cord injury.

Another study aimed at nerve regeneration was conducted by Quian et al. [102]. In this work, a boron nitride (BN)/PCL scaffold was implanted in mice. The scaffold presented no cytotoxic effects, and the nerve was reconnected 18 weeks post-implantation. The immune response derived from the inhibitory surrounds of the central nervous system was suppressed due to the ROS mediating role of this 2DnMat. An extensive explanation can be found in the original paper.

### 2.5. Potential Clinical Use

PDT is accomplished by activating ROS-generating photosensitizers (PS) through radiation exposure. Despite the encouraging prospects of pre-clinical and clinical trials using PDT for cancer treatment, most PS have limited solubility and tend to aggregate due to their hydrophobic nature and lack adequate tumor targeting capabilities, which lowers their treatment efficiency and can lead to long-lasting phototoxicity [123]. FDA-approved PS can be modified with 2DnMat to improve their stability and anti-cancer properties further. As mentioned throughout this review, 2DnMat are easily functionalized due to their high surface-to-volume area, are overall biocompatible and biodegradable, and are generally hydrophilic or can be modified to achieve ideal stability in biological media. These are properties that can improve the drawbacks currently observed with PS. Since nanostructured materials ranging from 100 to 200 nm are considered to be the optimal size to bypass clearance and achieve the enhanced permeability and retention (EPR) effect, two-dimensional nanomaterials can also greatly enhance tumor targeting [124].

PTT shares the same advantages as PDT for cancer treatment as they are non-invasive and selective treatments. In PTT, the heat generated by radiation-absorbing materials is used to induce cancer cell death. Pre-clinical trials have demonstrated the anti-cancer properties of PTT, including this review, where extensive research using 2DnMat for PTT has been investigated. The current clinical trials using PTT are based on laser systems alone that excite endogenous tissue chromophores [25].

Before introducing 2DnMat in biomedical systems, it is critical to assess their biosafety. Systematic and thorough in vitro and in vivo testing, both short and long-term, are the first steps prior to human clinical trials. Bioaccumulation in the major organs or tissues, impurities/contamination, dosage, ROS generation, and particle size are some of the biggest concerns regarding the use of nano-scaled materials [125,126,127,128]. Graphene has been widely researched for biomedical applications and has demonstrated great properties for its use in cancer treatment, despite controversial results regarding its biosafety [126,129]. BP has been demonstrated to have higher biocompatibility than graphene, especially when incorporated in biopolymers or surface-functionalized [130]. The MXene biosafety has been studied using the zebrafish embryo model [131]. The results suggested that the material did not present significant cytotoxic effects. TMDs preliminary studies have also demonstrated superior biocompatibility compared with graphene [132]. However, that same study also highlighted the urgent need to conduct further research on these materials’ biosafety. Surface functionalization can improve the biocompatibility of 2DnMat by enhancing their solubility and stability in physiological environments, consequently decreasing their bioaccumulation [127,132]. As discussed throughout this review, 2DnMat does not induce significant cytotoxic responses in vitro or in vivo models. No study observed damage/abnormalities in the major organs of the in vivo models. 2DnMat excretion from the organism was also successfully observed. Currently, most of the conducted research only carries short-term experiments, which is not enough to infer the nanomaterial’s biosafety. However, one of the discussed works explored the long-term biosafety of BP [80], opening doors for further research on the prolonged physiological effects of 2DnMat.

Despite the complexity of clinical translation from laboratory-scale testing, the prospects of seeing the use of 2DnMat in clinical trials for treatment based on phototherapies are optimistic. There are 50 nanopharmaceuticals currently employed in clinical practice [133]. Moreover, some photothermal nanomaterials such as silica–gold and silica–gold iron-bearing nanoparticles, are already FDA-approved for clinical use and trials [134]. The recent approval of using iron oxide nanoparticles for brain tumor treatment (NanoTherm^®^) is also paving a bright future regarding the use of nanomaterials in clinical settings. Even though major improvements have been made in the size consistency and yield in production methods, efforts are still being conducted by the scientific community to further improve the scalability to industrial-grade fabrication of 2DnMat which allows their use in clinical applications beyond current laboratory testing [135,136].

## 3. Summary and Conclusions

This review presented an overview of potential applications of emerging two-dimensional nanomaterials-based polymeric composites in the biomedical field. Two-dimensional nanomaterials have been widely researched and employed in energy storage and optoelectronics applications. It was only recently that the enticing properties of these materials enriched the biomedical field with their capability of revolutionizing conventional medical treatments. Furthermore, it was briefly discussed how these emerging 2D nanomaterials compare with the already established graphene. Higher biocompatibilities and electrical conductivities are some of the superior aspects that surpass graphene. The discussed 2DnMat present good biocompatibility, with BP having the highest biocompatibility and biodegradability properties in the 2DnMat family. Nonetheless, all the materials benefit from surface functionalization to improve their stability and biocompatibility.

The photothermal conversion efficiency and the surface-to-volume ratio are the main aspects that make 2DnMat attractive for biomedical applications. Combining these with already established hydrogels and scaffolds, has been shown to be highly beneficial. For the most part, the mechanical and electrical properties and overall performance of the polymeric platforms were further improved, alongside increased cell growth and proliferation provided by the additional biological cues given by the 2DnMat.

From the collected data, the polymeric composites are mostly used for photothermal cancer treatment, with or without chemotherapeutic drug delivery. Wound healing applications combined with the antibacterial properties of the 2D nanomaterials, are the second most studied application. Lastly, the polymeric composites were proposed to be used for biosensing applications. By far, the most researched material is the MXene Ti_3_C_2_/Ti_3_C_2_T_x_, closely followed by BP. However, due to the exceptional photothermal conversion efficiency of BP analogues, such as SiP and GeP, it is possible that these materials could become a focus for PTT applications.

The on-demand drug delivery explored in this review is achieved through the photothermal conversion ability of 2D nanomaterials. The drug could be adsorbed on the 2DnMat surface or encapsulated in the polymer. In either scenario, the increased temperatures from the photothermal conversion promote desorption from the nanomaterial’s surface or increase the interchain free volume that enables drug diffusion. It can also be a combination of both mechanisms. Furthermore, the photothermal conversion can also generate enough heat to induce tissue hyperthermia. By having the same heat-generating mechanism for both drug delivery and PTT, the composites can be employed in a dual-mode treatment. In vivo studies conducted on tumor-bearing mice revealed negligible tumor recurrence rate, demonstrating the effectiveness of this combined treatment. Moreover, this approach can bypass the debilitating side effects of intravenous chemotherapy.

Another multifunctional approach for the 2D nanomaterial-based polymeric composite approach was cancer treatment combined with tissue regeneration, particularly in the context of bone tissue. 2D nanomaterials demonstrated enhanced osteogenic capacity. Combining these materials with osteogenic-inducing scaffolds, like BG, allowed for a more efficient recuperation.

Open wounds are highly susceptible to infection, and their treatment can be quite challenging due to the rise of multidrug-resistant microorganisms. The antimicrobial properties of the 2DnMat dispersion as colloids are mainly accredited to the nano-knife structure of the nanosheets. 2DnMat sharp edges cut microorganisms’ cell membranes, which subsequently leads to their death. Regarding 2DnMat free-standing films or composites, NIR irradiation is able to increase local temperature leading to the destruction of microbes. By using 2DnMat, microbes’ elimination can be accomplished without using drugs. In addition, 2D nanomaterials facilitate angiogenesis by supplying the cells with biological cues that induce growth and proliferation. Therefore, a faster healing process can be achieved by incorporating 2DnMat in wound dressing hydrogels/scaffolds.

One of the main advantages of incorporating 2DnMat in polymeric scaffolds/hydrogels is an efficient and localized treatment. This way, current invasive treatments that inevitably cause the patient discomfort can be avoided without compromising the outcome.

Lastly, the next-generation wearable sensors for acquiring physiological signals were also explored in this review. They could potentially surpass graphene-based biosensors due to their greater hydrophilicity that allows better integration in physiological environments, enhanced electrical conductivity, and their potential ability to promote more efficient self-healing behavior of composites. Commercializing these composite-based sensors could lead to a new era of personalized healthcare and medicine.

Figure 8 briefly summarizes the steps and challenges of employing 2DnMat beyond in vitro testing. Despite the need to improve 2DnMat production techniques to have on-demand material for clinical testing and systematically study the long-term effects of 2DnMat, excellent prospects for these composites for biomedical applications are to be seen. The pre-clinical trials using these materials demonstrate their suitability for biomedical applications. The potential to revolutionize the current medical approaches is evident, especially in cancer treatment, tissue regeneration, and wound healing fields.

## Figures and Tables

**Figure 1 polymers-14-01464-f001:**
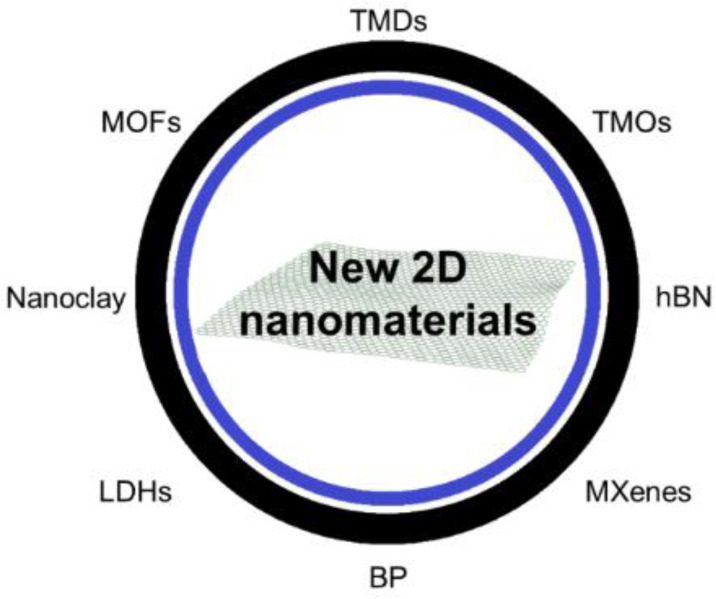
Summary of new 2D nanomaterials beyond graphene. These include transition metal dichalcogenides (TMDs); transition metal oxides (TMOs); hexagonal boron nitride (hBN); transition metal carbides, nitrides and carbonitrides (MXenes); black phosphorus (BP); layered double hydroxides (LDHs); nanoclay; 2D metal-organic frameworks (MOFs).

**Figure 3 polymers-14-01464-f003:**
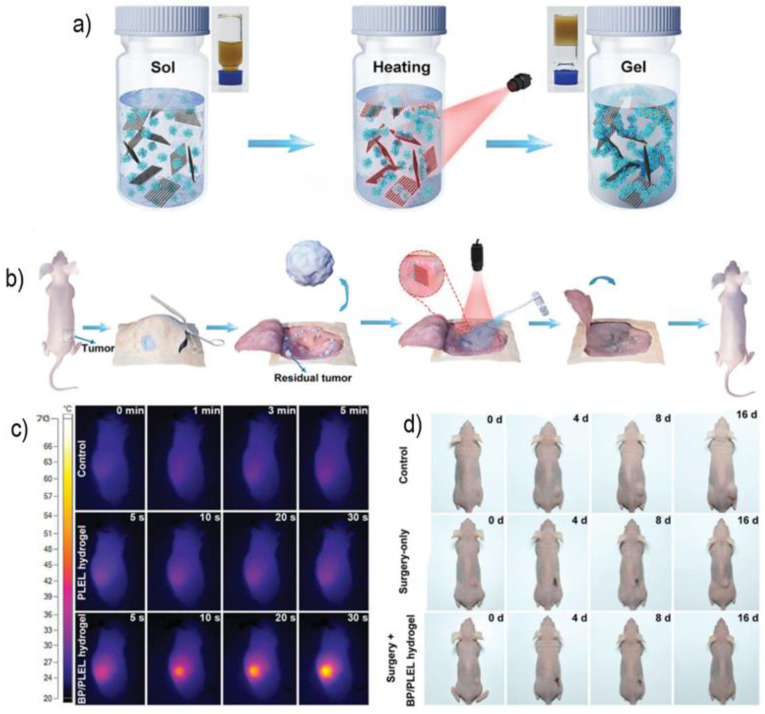
(**a**) Schematic of the BP/PLEL hydrogel gelation triggered by NIR radiation; (**b**) Schematic illustration of the post-surgical application of the BP/PLEL hydrogel. NIR induces gelation and promotes PTT of residual tumor while providing the site with an antibacterial platform; (**c**) time dependent thermographic map of the control, pristine PLEL, and BP/PLEL with NIR radiation; (**d**) photographs of the treatment progress throughout 16 days for control, surgery-only, and surgery + BP/PLEL with NIR hydrogel. No recurrence for the composite. Adapted from [73] © 2018 The Authors. Published by WILEY-VCH Verlag GmbH & Co. KGaA, Weinheim.

**Figure 4 polymers-14-01464-f004:**
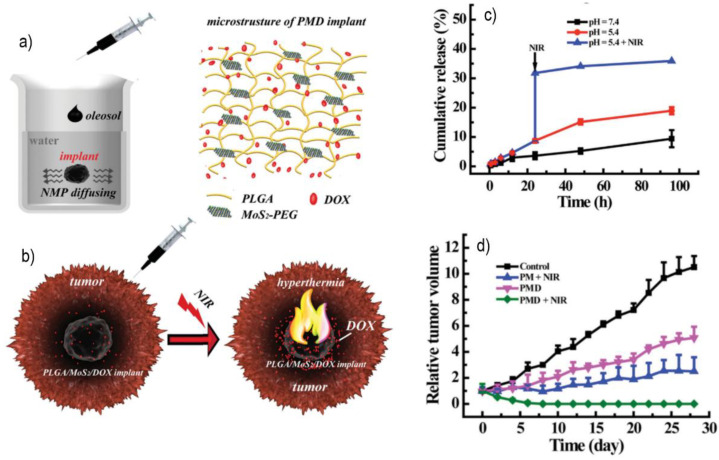
(**a**) Schematic representation of the solidification process when in contact with water and the microstructure of the implant; (**b**) diagram of the implant application in vivo, injection of the implant on the tumor which induces tissue hyperthermia upon NIR exposure; (**c**) DOX release profile under three different conditions (pH = 7.4; pH = 5.4; pH = 5.4 + NIR); (**d**) tumor volume under different treatments (control; PLGA/MoS_2_ (PM); PLGA/MoS_2_/DOX oleosol (PMD); PMD + NIR). Reproduced with permission from [81] © 2015 WILEY-VCH Verlag GmbH & Co. KGaA, Weinheim.

**Figure 5 polymers-14-01464-f005:**
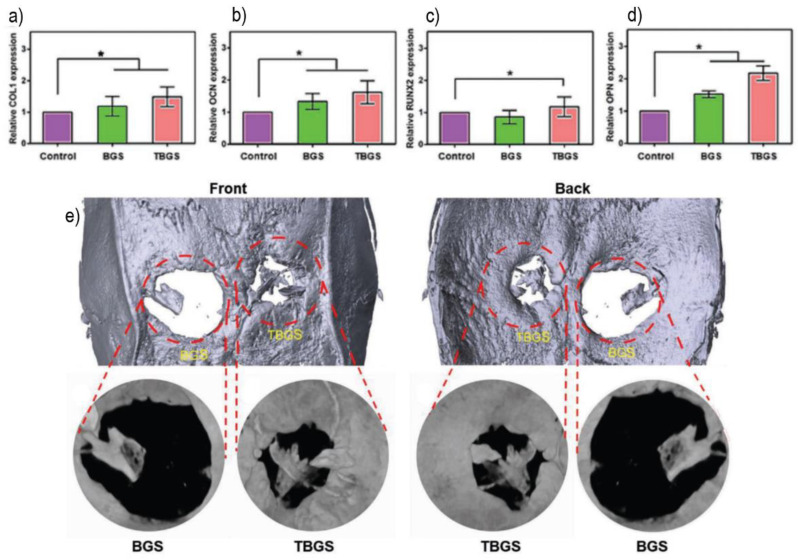
Osteogenic gene expression of (**a**) COL-1; (**b**) OCN; (**c**) RUNX2; (**d**) OPN in control, bioglass scaffolds (BGS) and BG/Ti_3_C_2_ scaffolds (TBGS). * represents a *p* < 0.05 significance. (**e**) in vivo osteogenic performance of BGS and TBGS scaffolds at 24 w post-scaffold implantation. Adapted from [92] © 2019 The Authors. Published by WILEY-VCH Verlag GmbH & Co. KGaA, Weinheim.

**Figure 6 polymers-14-01464-f006:**
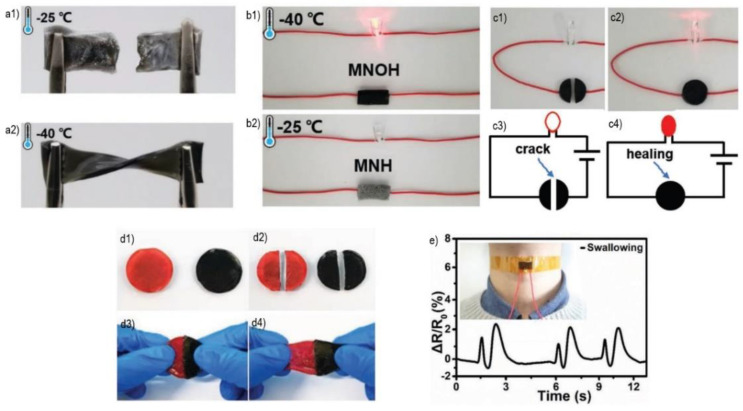
Comprehensive overview of the organohydrogel performance. (**a**) low temperature behavior of the composite pre-EG (ethylene glycol) (**a1**) and post-EG (**a2**); (**b**) electrical conductivity performance of the organohydrogel at low temperatures post-EG (**b1**) and pre-EG (**b2**); (**c**) conducting performance of the post-EG organohydrogel cut in half (**c1**), after self-healing (**c2**), and the corresponding schemes (**c3**) and (**c4**) respectively; (**d**) self-healing behavior of the post-EG organohydrogel. (**d1**) Two pieces of the hydrogel (black—original; red—dyed with rhodamine B). (**d2**) Each piece was cut in half. (**d3**) Self-healed hydrogel with one piece from each (black and red). (**d4**) Stretching behavior after self-healing; (**e**) real-time detection of throat swallowing movements. Reproduced with permission from [88] © 2019 WILEY-VCH Verlag GmbH & Co. KGaA, Weinheim.

**Figure 7 polymers-14-01464-f007:**
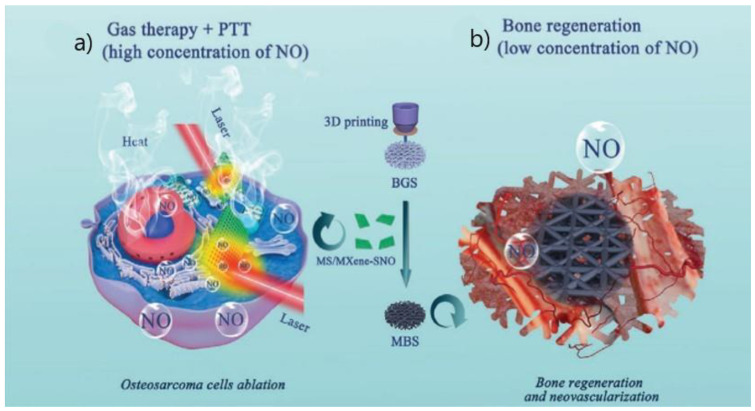
Schematic illustration of the multifunctional therapeutic platform. (**a**) NO gas release triggered by NIR radiation combined with PTT destroys osteosarcoma cells while (**b**) slow release of NO post-PTT stimulates bone regeneration. Reproduced with permission from [99] © 2020 WILEY-VCH Verlag GmbH & Co. KGaA, Weinheim.

**Figure 8 polymers-14-01464-f008:**
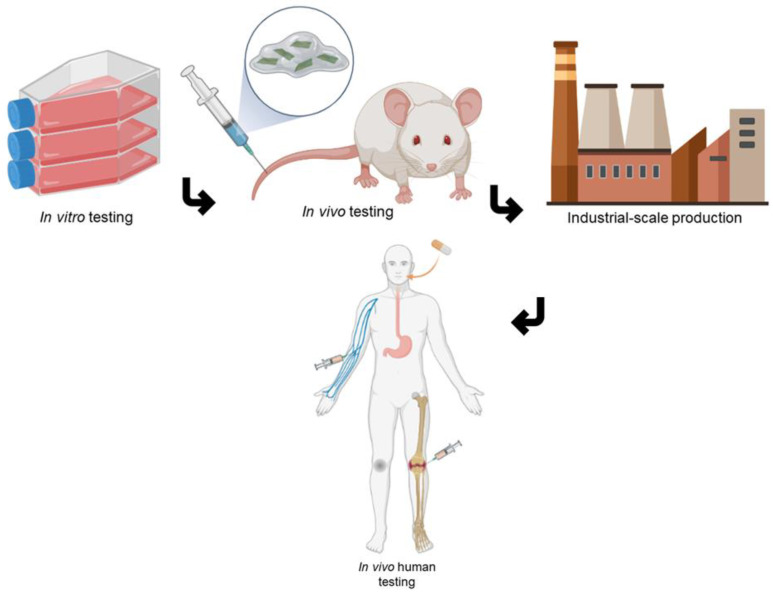
Brief schematic illustration of the steps and challenges of incorporating 2DnMat into systems for biomedical applications. Firstly, in vitro testing to infer on the biocompatibility of the nanomaterials and their interaction with cells must be conducted. The second logical step is to proceed to in vivo testing and evaluate the biocompatibility and therapeutic effects on live models. The ability to mass produce the nanomaterials is imperative for their commercialization and large-scale use. Lastly, human testing is crucial for the evaluation of the nanomaterials therapeutic effect and its safety on humans. Created in BioRender.com.

**Table 1 polymers-14-01464-t001:** Two-dimensional nanomaterials-based polymeric composites for biomedical applications, their specific functions, and general outcomes (↑ : increase; ↓ : decrease).

2DnMat	Composite	Composite Preparation	Application	Outcomes	Ref.
BP	PVA/pBP Hydrogel	pBP solution added to PVA solution followed by the freezing/thawing method.	NIR-Responsive Drug Release	↑ mechanical properties ↑ drug loading level (LL) with ↑ pBP% ↑ drug release with NIR exposure No cytotoxic effects observed with 3T3 fibroblasts	[72]
	PLEL/BP Hydrogel	BP nanosheets were dispersed in a PLEL solution by sonication.	Sprayable Gel for PTT	No cytotoxic effects observed for hMSCs, L929 and HeLa cells. In vivo biocompatibility: no histological abnormalities ↓ tumor recurrence ↓ >99.5% of *S. Aureus* reduction with NIR irradiation	[73]
	Cellulose/BP Hydrogel	Celullose, BP, and epichlorohydrin solutions were mixed. The solution was cross-linked. Dialysis was performed to removed excess reagents.	PTT	↑ mechanical properties with ↑ BP% No cytotoxic effects observed for B16, SMMC-7721 and J774.1 cells. In vivo biocompatibility: no histological abnormalities or increase of inflammatory cytokines levels ↓ tumor volume	[74]
	Agarose/BP@PEG Hydrogel, loaded with DOX	BP@PEG nanosheets, agarose aqueous solution and DOX were mixed and rapidly cooled.	Drug Delivery Induced by PTT	No cytotoxic effects observed for MDA-MB-231, A549, HeLa and B16 cells ↑ drug release with NIR exposure ↓ MDA-MB-231 cells with NIR induced drug release In vivo biocompatibility: no histological abnormalities ↓ tumor volume with combination of drug delivery + PTT	[75]
	BP/PEA/GelMA Hydrogel	A GelMA and PEA solution was submitted to photopolymerization. BP nanosheets were added to the hydrogel, followed by UV irradiation.	Bone Regeneration	↑ water-absorption capacity No cytotoxic effects were observed for hDPSCs cells ↑ mineralization and ↑ osteogenic differentiation of hDPSCs In vivo: Newly formed vessels were detectable in 4 weeks and after 12 weeks, the bone defect was completely repaired	[76]
	Fibrin/BP Gel	Fibrinogen solutions with BP nanosheets were mixed with thrombin through spraying.	Diabetic Ulcer Treatment + Analgesic + Antibacterial PTT	↑ gelation time with ↑ BP% ↑ proliferation and differentiation HUVECs ↓ ~94.3% of bacteria with NIR ↑ drug release triggered by NIR ↑ wound healing in vivo with the combination of composite + drug delivery + NIR ↓ 50% wound area in 5.7 days	[77]
	BG/BP 3D-printed Scaffold	BG scaffolds were 3D-printed and subsequently soaked in a BP absolute ethyl alcohol solution.	PTT + Bone Regeneration	↑ proliferation of Saos-2 cells ↓ tumor recurrence ↑ osteogenic differentiation of hBMSCs ↑ in vivo bone tissue formation.	[78]
	GelMA/BP@PDA Hydrogel	BP@PDA and GelMA solutions were sonicated until homogeneous, followed by UV irradiation.	MSCs Differentiation	↑ BP@PDA ↓ the swelling ratio of the hydrogel ↓ impedance for ↑ BP% ↓ degradation rate, presumably due to the functionalization with PDA that stabilizes the network Electrical stimulation ↑ MSCs proliferation ↑ neuronal gene expression in vitro Contrary to in vitro assays, the degradation of the composite is faster in vivo.	[79]
	BP/PCL Nanoscaffold	BP nanoplates were incorporated in a PCL dicloromethane solution. The solution was sprayed onto a conduit shaped mold.	Neural Regeneration	↑ electrical conductivity with ↑ %BP In vivo biocompatibility: no histological abnormalities and no increase of apoptotic cell markers No increase of blood biochemical parameters 6 months post-implatation ↑ angiogenesis	[80]
MoS_2_	PLGA/MoS_2_@PEG/DOX Injectable Implant	PLGA was dispersed in NMP, MoS_2_ was dispersed in the PLGA/NMP solution and DOX was dissolved in the PLGA/MoS_2_ dispersion.	Drug Delivery + PTT	No cytotoxic effects were observed for L929 cells. No increased blood coagulation 95% of DOX drug was loaded onto PLGA ↑ drug release with NIR exposure ↓ tumor volume actively reduced with NIR	[81]
	PAN/MoS_2_ Nanofibers	PAN was added to a MoS_2_/N,N-dimethylformamide solution and electrospun.	Composite Effects on BMSCs	↑ %MoS_2_ ↑ nanofiber surface roughness The fibers presented very low cytotoxicity even at 40% MoS_2_ and high rates of cell attachment 40% MoS_2_ ↑ osteogenic differentiation	[82]
Ti_3_C_2_T_x_	PLA/Ti_3_C_2_T_x_@OTES Membrane	Solvent casting was used to embed Ti_3_C_2_T_x_@OTES in PLA.	Bone Regeneration	↑ mechanical properties of the membrane are increased prior to saturation of the filler ↑ MC3T3-E1 cell adhesion and ↑ proliferation ↑ Osteogenic differentiation	[83]
	PNIPAM/Ti_3_C_2_T_x_ Hydrogel	PNIPAM and a cross-linker were added to a Ti_3_C_2_T_x_ solution and purged, followed by the addition of a polymerization accelerator.	PTT	MXene did not disrupt the hydrogel network Good photothermal stability Promising prospects for biomedical and drug delivery	[84]
	PLA/Ti_3_C_2_T_x_ Nanofibers	Electrospun PLA was immersed in a Ti_3_C_2_T_x_ solution.	Antibacterial	↑ roughness of the nanofibers No cytotoxic effects were observed with U2OS cells ↓ adhesion of *S. aureus* bacteria	[85]
	Chitosan/Ti_3_C_2_T_x_ Nanofibers	Ti_3_C_2_T_x_ was loaded onto a chitosan solution and electrospun.	Antibacterial Wound Dressing	Stable electrospinning GA crosslinked composite fibers exhibited a bacterial reduction of 95% for *E. Coli* and 62% for *S. aureus* No cytotoxic effects were observed with HeLa cells	[68]
	F127-PEI-OSA/Ti_3_C_2_T_x_@CeO_2_ Hydrogel	A solution of F127-PEI, OSA, and Ti3C2Tx@CeO2 was prepared was kept at 37 °C.	Multifunctional Wound Healing Scaffold	No cytotoxic effects were observed with L929 cells ↓ ROS with the addition of Ti_3_C_2_T_x_@CeO_2_ ↑ fibroblast proliferation with composite + ES ↑ healing on in vitro scratch assay: within 24 h, composite + ES reduced the unhealed portion by 73.6% ↓ 100% of bacterial colonies ↑ wound healing and antibacterial in vivo ↑ anti-inflammatory cytokines In vivo biocompatibility: no histological abnormalities	[86]
	PGE/HCHO/Ti_3_C_2_T_x_@PDA Scaffold	Ti_3_C_2_T_x_@PDA and PEG solutions were mixed in a HCHO solution and vortexed until homogenous.	Multifunctional Antibacterial Wound Healing	L929 cells adhered to the scaffolds ↓ bacteria growth of 98.6% for *E. coli*, 99.9% for *S. aureus* and 99.03% for *MRSA* ↓ coagulation time in vivo In vivo biocompatibility: no histological abnormalities ↑ wound healing observed by ↑ α-actin, COL III, and VEGF	[87]
	PAAm-PVA/Ti_3_C_2_T_x_ Hydrogel	The individual components that comprise the hydrogel were mixed in a aqueous solution with an initiator, followed by the addition of borax until a gel is formed.	Biosensor	↑ hydrogel conductivity ↑ antifreezing properties ↑ sensitivity to monitor human activities	[88]
	PVA/Ti_3_C_2_T_x_ Hydrogel	A Ti_3_C_2_T_x_ solution was mixed with a PVA solution, followed by the addition of borax.	Biosensor, Electronic skin	↑ hydrogel stretchability self-healing ability remains at 97.4% after 5 cycles of cutting ↑ sensitivity, it detects swallowing and finger motion	[89]
	PVDF-TrFE/Ti_3_C_2_T_x_ Nanofibers	Ti_3_C_2_T_x_ was added to a PVDF-TrFE solution and electrospun.	Biosensor for Physiological Signal Acquisition	The sensor showed good capabilities in recognizing pulse signals in the wrist, breathing, and promising results for future aiding of Parkinson’s diagnose by measuring unnoticeable resting tremor in hands	[90]
Ti_3_C_2_	Ti_3_C_2_/Honey/Chitosan Hydrogel	Ti_3_C_2_ was added to a chitosan hydrogel solution, followed by the addition of honey, β-glycerophosphate and hydroxyethyl cellulose.	Biomedical Applications	Good swelling ability, biodegradable, and self-healing No cytotoxic effects were observed for MSCs and iPSCs cells	[91]
	BG/Ti_3_C_2_ 3D-Printed Scaffolds	3D-printed scaffolds were soaked in Ti_3_C_2_ aqueous solution.	PTT + Bone Regeneration	No cytotoxic effects were observed for Saos-2 cells ↑ adhesion and ↑ proliferation of hBMSCs ↑ osteogenic capability In vivo biocompatibility: no histological abnormalities ↑ differentiation from ↑ of COL I, RUNX2, OCN and OPN gene expression ↑ bone regeneration; ↑ tissue calcification	[92]
	PCL/Ti_3_C_2_ Electrospun Scaffolds	PCL was added to Ti_3_C_2_ was dispersed on a dimethylformamide and chloroform solution and electrospun.	Biomedical Applications—study	↑ fiber diameter with ↑ MXene content ↑ biomineralization in vitro The scaffold is more biocompatible for MC3T3-E1 than NIH-3T3 cells Promising prospects for wound healing, bone TE and cancer therapy	[93]
	Cellulose/Ti_3_C_2_ Hydrogel	The same method as described in [74].	PTT + Drug Release	No cytotoxic effects were observed for HepAl-6, SMMC-7721, HepG2, U-118MG and U-251MG cells. NIR irradiation yielded in ~100% killing efficiency of tumor cells In vivo biocompatibility: no histological abnormalities Dual modal PTT/chemo was successful in vivo as it completely eliminated tumor cells	[94]
	PAM/Ti_3_C_2_ Hydrogel	The hydrogel was prepared using a free radical polymerization method. An aqueous Ti_3_C_2_ solution was mixed with acrylamide and an initiator to initiate polymerization.	Drug Release	↑ mechanical properties of the hydrogel ↑ drug loads and ↑ drug release	[95]
	PLLA-PHA/Ti_3_C_2_ Nanofibers	PLLA and PHA were added to a Ti_3_C_2_ dichloromethane/dimethylformamide solution.	Tissue Engineering	↑ adhesion and slightly ↑ proliferation of BMSCs Cells could grow on both scaffolds (w/ and w/o MXene) but on the pristine nanofibers the cells presented a contraction state. The composite nanofibers present, overall, a positive role on BMSCs growth and can enhance osteogenic ability	[96]
	PAN-PVP/Ti_3_C_2_@PAAV Fibrous Nanobelts	PAN and PVP were added to a Ti_3_C_2_ dimethylformamide solution and electrospun. The nanofibers were soaked in a PAAV aqueous solution.	Wound Healing + Drug delivery	↑ adhesion and ↑ proliferation of BMSCs ↑ vitamin E release with NIR radiation ↑ wound healing with NIR in vivo, displaying the advantage of vitamin E release	[97]
	PVA/AMX/Ti_3_C_2_ Nanofibrous Membrane	PVA was dissolved in a Ti_3_C_2_ aqueous solution, followed by the addition of AMX. The final solution was electrospun.	Wound Healing + Drug Delivery	No cytotoxic effects were observed for L929 cells ↑ AMX release with NIR radiation ↑ antibacterial properties with NIR: ↓ 96.1% for *E. coli* and ↓ 99.1% for *S. aureus* In vivo biocompatibility: no histological abnormalities with NIR↑ wound healing rates and neglectable inflammation	[98]
Nb_2_C	BG/Nb_2_C@Silica 3D-printed Scaffold	A mesoporous silica layer was coated onto Nb_2_C nanosheets. 3D-printed BG scaffolds were soaked on a Nb_2_C@Silica solution.	Bone Regeneration	↑ NO release is with NIR radiation, maintaining a slow-release profile after No cytotoxic effects were observed for Saos-2 cells. In vivo tests were carried out in mice. NO release + photothermal therapy yielded the better results. ↑ adhesion and ↑ proliferation for hBMSCs ↑ osteoindcution properties in vivo ↑ increased calcified tissue	[99]
SiP	GelMA-PEGDA/SiP@AC 3D-printed Hydrogel	GelMA, PEGDA, and SiP@AC were mixed in a phosphate-buffered saline solution and 3D-printed.	Bone Regeneration	↑ release of P ions No cytotoxic effects below 0.5% SiP@AC ↑ ALP expression ↑ mineralized nodules ↑ osteogenic differentiation markers (Opn, Runx2 and Col-I) ↑ angiogenic genes (VEGF and bFGF) ↑ in vivo bone growth In vivo biocompatibility: no histological abnormalities	[100]
GeP	HA-DA/GeP@PDA Injectable Hydrogel	HA-DA and GeP@PDA aqueous solutions were mixed. Horseradish peroxidase was added as an initiator for the cross-link of the hydrogel.	Spinal Cord Injury Repair	↓ swelling ratio but ↑ conductivity No cytotoxic effects were observed for concentrations below 0.5% of GeP@PDA ↑ NSCs differentiation ↑ coordination movements in vivo ↓ spinal cord cavity ↑ anti-inflammatory factor IL-10 and ↓ TNF-α 6 weeks post-surgery ↑ levels of CD31-labeled vascular endothelial cells ↓ invasion of lesion area and ↑ secretion of vascular endothelial growth factor ↑ angiogenesis enhanced by P element released from the GeP nanosheets. P ↑ Akt protein, MMP-2 and bFGF expression, which enhances new blood vessel formation	[101]
Boron Nitride (BN)	BN/PCL Scaffold	BN nanosheets were added to a PLC/dichloromethane solution and sprayed onto a rotatory mould.	Nerve Regeneration	↑ mechanical properties with ↑ BN% No cytotoxic effects were observed for RSC96 cells ↓ immune response In vivo biocompatibility: no histological abnormalities ↑ S100 and Tuj1	[102]

AC: acryloyl chloride; AMX: amoxicillin; bFGF: basic fibroblast growth factor; BG: bioglass; BMSCs: human bone marrow stromal cells; BP: Black Phosphorus; DA: dopamine; ES: electrical stimulation; GA: glutaraldehyde; GelMA: gelatin methacrylamide; GeP: germanium phosphideHA: hyaluronic acid; hBMSCs: human bone marrow stromal cells; HCHO: oxidized hyaluronic acid; hDPSCs: human dental pulp stem cells; hMSCs: human mesenchymal stem cells; HUVECs: human umbilical vein endothelial cells; MoS_2_: molybdenum disulphide; MSCs: mesenchymal stem cells; Nb_2_C: niobium carbide; NIR: near infrared; NMP: N-methylpyrrolidone; NO: nitric oxide; OSA:oxidized sodium alginate; OTES: n-octyltriethoxysilane; PAAm: polyacrylamide; PAAV: P(AAm-co-AN-CO-VIm); PAN: polyacrylonitrile; pBP: modified BP with poly(dopamine); PCL: polycaprolactone; PDA: poly(dopamine); PEA: poly(ester amide); PEG: poly(ethylene glycol); DOX: doxorubicin; PEGDA: Poly(othylene glycol) diacrylate; PEI: polyethylenimine; PGE: poly(glycerol-ethylenimine); PHA: polyhydroxyalkanoates; PLA: poly(lactic acid); PLEL: poly(d,l-lactide)-poly(ethylene glycol)-poly(d,l-lactide); PLGA: from poly(lactic-co-glycolic acid); PLLA: poly-L-lactic acid; PNIPAM: poly(N-isopropylacrylamide); PTT: photothermal therapy; PVA: poly(vinyl alcohol); PVA: polyvinyl alcohol;PVDF: poly(vinylidene) fluoride; PVP: polyvinylpyrrolidone; ROS: reactive oxygen species; SiP: silicon phosphide; TE: tissue engineering; Ti_3_C_2_(T_x_): titanium carbide; TrFE: trifluoroethylene; UV: ultra-violet; VEGF: vascular endothelial growth factor.

## Data Availability

Not applicable.

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
