# Peer review of "New Polymeric Composites Based on Two-Dimensional Nanomaterials for Biomedical Applications"

_polymers, 2022, doi:10.3390/polym14071464_

Round 1

Reviewer 1 Report

Reviewer’s comments:

The manuscript entitled ‘New Polymeric Composites based on Two-Dimensional Nanomaterials for Biomedical Applications’ has been peer-reviewed. The manuscript demonstrates the biomedical applications of 2D nanomaterials. We have provided the following comments to improve the manuscript.

Minor concerns:

1) Abstract needs to be reframed. Please revise the following statements.

One of the ongoing hot topics in the scientific community is two-dimensional nanomaterials. Their unique properties make them attractive to a wide array of applications, including biomedical applications.

2) The authors have demonstrated the most recent works employing polymeric composites for biomedical applications. However, the novelty of the present work was not indicated properly showing the difference from the related reviews, published already.

3) Table 1. Some phrases are placed without each word capitalization. Please maintain homogeneity throughout the table content presentation.

Sprayable gel for PTT, Composite effects on BMSCs, Biosensor for physiological signal acquisition, BG/Nb2C@Silica 3D-printed scaffold, Bone regeneration.

4) Please write the molecular structure properly. See the table footnotes (lines 180-190)

Ti3C2(Tx): titanium carbide

MoS2: molybdenum disulphide

Nb2C: niobium carbide

5) 2.4. Others

The authors can write as ‘Other 2D nanomaterials’ instead of ‘Others’.

6) In conclusion, the authors have simply summarized the manuscript content. We expect the possible challenges and future directions in the field. Please mention if any commercialized products based on 2D nanomaterials in the mentioned biomedical applications are available in the market.

Reviewer 2 Report

In this manuscript, the authors reviewed two-dimensional nanomaterials-based polymeric composites and their applications in biomedicine, including tissue regeneration, cancer phototherapy, antibacterial, drug delivery, and biosensing. This review provides meaningful guidance for the biomedical application of two-dimensional polymeric composites. The specific comments are as below:
1. The format of references could be improved.
2. The design of "Introduction" still needs to be improved, this section is too lengthy and it is necessary to emphasize the innovative aspects of the review.
3. English needs to be polished.
4. In section 2, it is required to improve the logical and structural organization of the content.

Reviewer 3 Report

The authors have done a systematic and detailed review on recently reported polymeric composites based on 2D nano materials for biomedical applications .Their coverage of MXenes was really good.   If the authors have included separate section for biosensing applications it would be more appealing.   Since the authors are focusing on 2D materials, it would be better if the authors could discuss about the current and the future aspects of Xenes like Borophene composites for Biomedical applications.   Further, it is important to include the future perspective in this field based on their review.

Round 2

Reviewer 2 Report

In this manuscript, the authors reviewed recent two-dimensional nanomaterials-based polymeric composites and their applications in biomedicine, including drug delivery, wound healing, tissue engineering, gas therapy, and biosensing. This review is complete, detailed, and clear in structure. Based on the wide application of two-dimensional nanomaterials in the biomedical field, it is suggested to discuss the safety of nanomaterials.
